# Partial Correspondence between Host Plant-Related Differentiation and Symbiotic Bacterial Community in a Polyphagous Insect

**DOI:** 10.3390/ani14020283

**Published:** 2024-01-16

**Authors:** Zhentao Cheng, Qian Liu, Xiaolei Huang

**Affiliations:** State Key Laboratory of Ecological Pest Control for Fujian and Taiwan Crops, College of Plant Protection, Fujian Agriculture and Forestry University, Fuzhou 350002, China; zhentao.cheng0123@gmail.com (Z.C.); liuqian9502@163.com (Q.L.)

**Keywords:** herbivorous insect, host specialization, species diversification, symbiont community, *16S*

## Abstract

**Simple Summary:**

The host plants and the symbiotic relationship between bacteria and herbivorous insects play important roles in the population differentiation of phytophagous insects. However, current studies have paid limited attention to the connection between host-related differentiation and symbiotic bacterial communities in phytophagous insects. This study constructed phylogenetic trees using 58 *Aphis odinae* samples collected from different host plants and analyzed the symbiotic bacterial communities of samples from five commonly occurring host plants. The results indicate a partial correlation between differentiation related to host plants and the symbiotic bacterial communities in *A. odinae*, providing insights into the microevolutionary process between bacterial symbionts and insects.

**Abstract:**

Host plants play a vital role in insect population differentiation, while symbiotic associations between bacteria and insects are ubiquitous in nature. However, existing studies have given limited attention to the connection between host-related differentiation and symbiotic bacterial communities in phytophagous insects. In this study, we collected 58 samples of *Aphis odinae* from different host plants in southern China and constructed phylogenetic trees to investigate their differentiation in relation to host plants. We also selected aphid samples from the five most preferred host plants and analyzed their symbiotic bacterial composition using Illumina sequencing of the V3–V4 hypervariable region of the *16S* rRNA gene. The phylogeny and symbiotic bacterial community structure of *A. odinae* populations on different host plants showed that samples from *Triadica sebifera* (Euphorbiaceae) had a consistent presence of *Wolbachia* as the predominant secondary symbiont and suggested the possibility of undergoing differentiation. Conversely, although differentiation was observed in samples from *Rhus chinensis* (Anacardiaceae), no consistent presence of predominant secondary symbionts was found. Additionally, the samples from *Heptapleurum heptaphyllum* (Araliaceae) consistently carried *Serratia*, but no host differentiation was evident. In summary, this study reveals a partial correspondence between symbiotic bacterial communities and host-related differentiation in *A. odinae*. The findings contribute to our understanding of the microevolutionary influencing the macroevolutionary relationships between bacterial symbionts and phytophagous insects. The identification of specific symbionts associated with host-related differentiation provides valuable insights into the intricate dynamics of insect-bacteria interactions.

## 1. Introduction

Insect differentiation has long been thought to be associated with host plants [1], and numerous studies have confirmed the existence of genetically differentiated host races within insect species [2,3,4,5], particularly in polyphagous insects, known for their ability to feed on a wide range of host plants [6]. On the other hand, symbiotic associations between insects and bacteria are widespread in nature. The mutually beneficial relationship between aphids and many advantageous intracellular bacteria is considered a significant contributing factor to the ecological prosperity of aphids [7,8]. For instance, as the unbalanced plant phloem diet is unable to provide necessary nutrients [9], *Buchnera*, an intracellular bacterium, supplies aphids with essential nutrients and plays a crucial role in aphid nutritional metabolism [10,11]. The association of aphids and their primary symbiont, *Buchnera*, is a classic model of insects and their endosymbionts [12]. Additionally, aphids may also harbor one or several secondary or facultative symbionts for additional nutrient acquisition and other assistance [13]. Some secondary (or facultative) symbionts can have environmentally-dependent effects on their host aphids, such as expanding the hosts’ viable temperature ranges [14,15], affecting aphid reproduction [16,17], and enhancing their hosts’ resistance to parasitic wasps [18,19,20] and fungal pathogens [21,22,23].

Secondary symbionts can influence herbivorous insects by adapting to their host plants [24,25]. At the population level of aphids, a rise in plant species richness resulted in an augmentation of the diversity within the aphid symbiont community [26]. In different host populations of the same aphid species, variations in symbiotic communities may also exist. [27,28,29]. For instance, infection with *Regiella* (Pea Aphid U-type Symbionts, PAUS) can cause *Acyrthosiphon pisum* Harris to be unable to survive on *Medicago sativa* (alfalfa) but improve their fitness on *Trifolium repens* (white clover) [30,31,32,33], while *Ac. pisum* from *Lotus* harbors *Hamiltonella*, and those from *Cytisus*, *Pisum,* and *Vicia* associate with *Serratia* [27]. Similar variations in symbiont communities have been reported in other aphid species, such as *A. craccivora* Koch [34] and *A. gossypii* Glover [35]. However, most of the research did not further analyze the population differentiation of aphids on different host plants to explore the relationship between the host plant-related microevolutionary process and the symbiotic bacterial community. This study aims to address this gap by investigating the population differentiation of aphids on various host plants and exploring its potential correlation with symbiotic bacterial community composition.

*Aphis odinae* van der Goot, commonly known as the mango aphid, exhibits a wide distribution in the tropical and subtropical regions of the Old World [36]. As a polyphagous insect pest, *A. odinae* inflicts significant damage on various plant species, especially those in families such as Anacardiaceae (*Anacardium*, *Mangifera*, *Rhus*), Araliaceae (*Heptapleurum*, *Aralia*), Euphorbiaceae (*Triadica*), Pittosporaceae (*Pittosporum*), Rubiaceae (*Coffea*), and Rutaceae (*Citrus*) [37,38,39,40]. The ability to feed on various host plants makes *A. odinae* a good example for studying the differentiation of aphid populations on different hosts and the potential association between this differentiation and the symbiotic bacterial community.

The phylogenetic trees based on the *COI* gene of *A. odinae* from different host plants were constructed. The symbiotic bacterial communities of *A. odinae* collected from the most preferred host plants were detected by utilizing *16S* rRNA Illumina sequencing. The V3–V4 region is known for containing variable regions in the *16S* rRNA gene, enabling discrimination among closely related microbial species or strains while retaining sufficient conserved regions for primer binding; it is widely employed in research on symbiotic bacterial diversity and host-related population differentiation in aphids [35,41,42,43]. Through these methods, we aim to shed light on the relationship between the host-related microevolutionary process and symbiotic bacterial communities in polyphagous aphids.

## 2. Materials and Methods

### 2.1. Sample Collection

*A. odinae* is widespread in southern China, causing damage to various plants. In this study, we collected a total of 58 *A. odinae* samples from nine typical host plant species in southern China during the period from April 2015 to November 2017. We mainly selected sympatric samples to exclude the potential influence of geographic factors on differentiation patterns. Detailed information (including sample ID, host plant, host family, and collection locality) is provided in Table 1. Each sample consists of multiple individuals obtained from one independent aphid colony on a single leaf (or shoot). The number of individuals in each sample ranged from tens to hundreds, depending on the size of the colony. To preserve the specimen’s integrity and prevent DNA degradation, field-collected samples were promptly immersed in 95% ethanol and stored at −20 °C in the laboratory for subsequent molecular and morphological analyses.

### 2.2. COI Gene Amplification and Sequencing

Total genomic DNA was extracted from a whole individual from each sample with the DNeasy Blood & Tissue kit (QIAGEN) according to the manufacturer’s manual. The primers LepF (5′-ATTCAACCAATCATAAAGATATTGG-3′) and LepR (5′-TAAACTTCTGGATGTCCAAAAAATCA-3′) [44] were used herein to amplify the *COI* sequences. PCR was performed in 50 µL reactions containing 28.5 µL ddH_2_O, 8 µL dNTPs, 5 µL 10× PCR Buffer, 4 µL template DNA, 2 µL of each primer (10 µM), and 0.5 µL of TaKaRa LA Taq (5 U/µL) (TaKaRa Bio Inc., Otsu, Japan). PCR thermal regimes were as follows: a 5 min initial denaturation at 95 °C followed by 35 cycles of 95 °C for 1 min, 50 °C for 1 min, 72 °C for 2 min, and a 10 min final extension at 72 °C. The products of PCR were visualized by electrophoresis on a 1% agarose gel and then bidirectionally sequenced at Beijing Tsingke Biotech Co., Ltd. (Beijing, China).

### 2.3. Phylogenetic Analysis

The double-ended sequences obtained from Beijing Tsingke Biotech Co., Ltd. (Beijing, China) were checked and assembled using ContigExpress (Vector NTI Suite 6.0, InforMax Inc., Bethesda, MD, USA). The taxonomic assignment of the assembled sequences was verified according to the results of BLAST. The phylogenetic tree based on *COI* sequences includes 58 *A. odinae* samples collected from different host plants; two *A. gossypii* and two *A. fabae* sequences were chosen as outgroups (Appendix A). Multiple alignments were conducted using MAFFT version 7 [45] based on default setting. Maximum likelihood phylogenies were inferred using IQ-TREE v2.2.0 [46] under the TN+I+F model for 5000 ultrafast [47] bootstraps. Bayesian Inference phylogenies were inferred using MrBayes v3.2.7a [48] under GTR+I+F model (2 parallel runs, 2,000,000 generations), in which the initial 25% of sampled data were discarded as burn-in. ModelFinder v2.2.0 [49] was used to select the best-fit model using BIC criterion. To further explore genetic differences among the populations, the sequences of *A. odinae* were used to generate a haplotype file in DnaSP 5.0 [50], and then median-joining networks were constructed using Network 5 [51]. Genetic distances based on the Kimura two-parameter (K2P) model were calculated in the MEGA 7 software [52].

### 2.4. DNA Extraction for 16S rRNA

In order to explore the differences in symbiotic bacterial community of *A. odinae* colonies from different hosts, ten out of 58 samples from five of the most typical host plants (*Triadica sebifera*, *Mangifera indica*, *Rhus chinensis, Heptapleurum heptaphyllum*, and *Toxicodendron sylvestre*) were examined (Table 2).

One apterous adult aphid from each sample was washed three times with ultrapure water. Total genomic DNA was extracted from the whole individual with the DNeasy Blood & Tissue kit (QIAGEN) according to the manufacturer’s manual. The process of DNA extraction was carried out on an ultra-clean workbench to avoid contamination of samples by environmental DNA. The bacterial universal primers 8F (5′-AGAGTTTGATCCTGGCTCAG-3′) and 1492R (5′-GGTTACCTTGTTACGACTT-3′) [53] were used to confirm the success of DNA extractions. PCR amplifications were performed as follows: a 4 min initial denaturation at 94 °C followed by 30 cycles of 94 °C for 30 s, 65 °C for 40 s, 72 °C for 90 s, and a 10 min final extension at 72 °C. The PCR products were checked with 1% agarose gel electrophoreses, and the positive samples with bright bands about 1500 bp were kept at −20 °C for subsequent operations. To assure the accuracy of results, two negative control samples of deionized water were included in the DNA extraction procedure and PCR amplifications, respectively.

### 2.5. High-Throughput 16S rRNA Gene Amplification and Sequencing

The V3-V4 hypervariable region of *16S* rRNA gene of the 10 samples was amplified with the primers 338F (5′-ACTCCTACGGGAGGCAGCA-3′) and 806R (5′-GGACTACHVGGGTWTCTAAT-3′) [54]. A total of two PCR procedures were performed, where the first PCR reaction was used to amplify the target regions and the second one was used to add indices and adapter sequences. The final PCR products were checked and recovered using 1.8% agarose gel electrophoresis; the positive products were purified and homogenized to form a sequencing library. Finally, the library pool was submitted to an Illumina HiSeq 2500 sequencing system (Illumina, Inc., San Diego, CA, USA) at Biomarker Bioinformatics Technology, Co., Ltd., (Beijing, China).

### 2.6. Sequencing Data Analysis

Paired-end reads were merged into single raw reads using FLASH v. 1.2.11 [55] with a minimum overlap size of 10 bp. Raw reads were further trimmed to obtain clean tags by Trimmomatic v0.33 [56], ensuring >20 quality scores on a sliding window of 50 bp, and tags shorter than 300 bp were also filtered. The chimeras were identified and removed to obtain high-quality, clean tags using UCHIME v8.1 [57]. The denoised sequences with ≥97% similarity were clustered into operational taxonomic units (OTUs) using USEARCH v10.0 [58]. OTUs of which sequences accounted for less than 0.005% of the total sequences were discarded. The most abundant sequences in each OTU cluster were screened as representative sequence for further annotation. Classification of each OTU was performed using the RDP classifier v2.2 [59] based on the SILVA database (Release138.1) [60]. The taxonomic assignments of OTUs were then manually checked by BLAST against GenBank sequences.

### 2.7. Diversity Analysis

Alpha diversity indices (Chao1 and ACE indices measuring species richness, Shannon and Simpson indices measuring community diversity) and the coverage of library for each sample were calculated based on the OTU data set using Mothur v1.30 [61]. The larger alpha diversity indices indicate the greater diversity of communities within the samples. All samples of *A. odinae* were grouped according to host plant species (including five groups, Table 2). Significance tests of the Chao1 and ACE indices for aphid symbionts from different plant groups were performed using one-way ANOVA test. Shannon and Simpson indices were found to deviate from normality (*p* < 0.05, Shapiro–Wilk test); therefore, the non-parametric Kruskal–Wallis test was used to check for significant differences across five host plant species. All statistical tests were calculated with IBM Statistical Package for the Social Sciences (SPSS) v 24.0 (Chicago, IL, USA). The relative abundance of the top three dominant symbionts (*Buchnera*, *Serratia*, and *Wolbachia*) in each sample was estimated by normalizing the number of sequences assigned to those three genera, respectively, against the total number of sequences obtained for a given sample.

The dissimilarities of the bacterial communities between different host plant species were also assessed. Considering the presence/abundance of bacteria, the Bray–Curtis distances based on OTUs and Weighted Unifrac distances based on phylogenetic information of OTUs were used to quantify beta diversity [62]. The bacterial communities among five host groups were clustered using nonmetric multidimensional scaling (NMDS) [63] and constrained principal coordinate analysis (CPCoA) in the package veganv2.3.0 [64], and two-dimensional plots were created in the ggplot2 v3.1.1 [65] in the R v3.1.1 programming environment. When stress is less than 0.2, the NMDS analysis is reliable.

## 3. Results

### 3.1. Phylogenetic Relationships between A. odinae Populations on Different Host Plants

Both the ML tree and Bayesian tree inferred from the *COI* gene revealed two well-supported host plant clades (Figure 1). One clade comprised all samples of *A. odinae* populations feeding on *Tr. sebifera* (Euphorbiaceae), marked as the *Triadica*-Group. The other clade primarily consisted of samples collected from *R. chinensis* (Anacardiaceae) and was marked as the *Rhus*-Group. The remaining samples from various host plant populations appeared at the base of the phylogenetic trees, and their phylogenetic relationships lacked clear associations with the host plants.

Among the 58 *COI* sequences examined, a total of nine haplotypes were identified (Figure 2). Haplotype H1 contained the highest number of samples (23 sequences) and was associated with eight host plant species. Haplotype H2 corresponded to *Rhus*-Group in the phylogenetic tree and included most of the samples feeding on *R. chinensis*, with the remainder found in haplotype H1 and haplotype H3. The samples feeding on *R. chinensis* accounted for 91.7% (11 in 12) of the total in H2. All the samples feeding on *Tr. sebifera* were assigned to haplotype H3, which also included one sample collected from *R. chinensis* and one from *Toxicodendron vernicifluum*. Haplotype H3 corresponded to the *Triadica*-Group in the phylogenetic tree. The samples feeding on *To. vernicifluum* accounted for 90% (18 out of 20) of all samples in haplotype H3. Consequently, haplotype H2 and haplotype H3 showed significant differentiation from the other haplotypes. The mean genetic distance between haplotype H1 and haplotype H2 was 0.54%, and between haplotype H1 and haplotype H3, it was 0.18%. The mean distance between haplotype H2 and haplotype H3 was 0.36%. The greatest genetic mean distance was found between haplotype H2 and haplotype H4, which amounted to 1.45% (Appendix A).

### 3.2. Symbiotic Bacterial Composition of Different Populations of A. odinae

#### 3.2.1. Library Basic Statistics

After quality control, we obtained 771,541 raw reads (77,154 reads per sample). After sequence filtering and discarding the OTUs with a number of sequences <0.005%, a total of 298,922 reads (29,892 reads per sample) were obtained (Appendix A). The average length of the assembled paired sequences of the *16S* rRNA gene was 428 bp.

#### 3.2.2. OTU Clustering and Taxonomic Assignment

The high-quality reads were assigned to 51 OTUs, and the details of OTU distribution in all samples are shown in Appendix A. The number of OTUs in each sample is presented in Appendix A. The reads of the top 10 genera of symbiotic bacteria in all samples are presented in Appendix A. The raw sequences have been submitted to the NCBI Sequence Read Archive (accession number: PRJNA1008577).

At the genus level, among all the tested samples, the obligate endosymbiont *Buchnera* was the most abundant symbiotic bacteria (93.26% of the total reads) and was represented by five OTUs (OTU1, OTU105, OTU214, OTU216, and OTU218) (Appendix A). As the primary endosymbiont, *Buchnera* predominated in all samples, with relative abundances ranging from 81.95% to 99.64%. As the secondary endosymbionts, *Serratia* was mainly distributed in samples feeding on *H. heptaphyllum* with an average relative abundance of 15.01%; *Wolbachia* was distributed in samples feeding on *Tr. sebifera* with a relative abundance of 2.25% to 4.59%; one sample from *To. sylvestre* (HCX56: 17.81%); and one sample from *M. indica* (HCX51: 0.03%) (Appendix A). The relative abundances of the other seven symbiotic bacteria in the top 10, excluding *Buchnera*, *Serratia*, and *Wolbachia*, were less than 1% in all tested samples.

#### 3.2.3. Alpha Diversity

The coverage of the library for each *A. odinae* sample was more than 0.99, indicating that the sequencing depth was high, and almost all symbiotic bacteria in the samples were detected, which could well reflect the bacterial composition in the samples. The four alpha diversity indices of symbiotic bacterial communities were distributed across all aphid samples, ranging from 21.9 to 51.1 (M ± SD/with a mean of 37.4 ± 1.5) for the ACE index, from 20.5 to 47.3 for the Chao1 index, from 0.05 to 0.36 for the Simpson index, and from 0.23 to 0.91 for the Shannon index (Appendix A).

The symbiotic communities of the five host groups of *A. odinae* samples revealed statistical differences in the ACE (*p* = 0.020, ANOVA test, Table 3) and Chao1 (*p* = 0.001, ANOVA test, Table 3) diversity indices. However, the Mann-Whitney test showed no significant differences in the Simpson (*p* = 0.221) and Shannon (*p* = 0.221) diversity indices.

#### 3.2.4. Beta Diversity

Beta diversity was used to explore the relations between host plants and symbiotic bacterial composition in the 10 *A. odinae* samples (Figure 3). The NMDS and PCoA analyses did not show a strong correlation between the symbiotic bacterial composition of *A. odinae* and their host plants because the samples were not separated into different clusters according to their host plants. Only the *Mangifera*-Group (host plant: *M. indica*) showed higher similarities in the symbiotic bacterial composition among samples.

## 4. Discussion

Host plants, serving as food and shelter, play a crucial role in aphid differentiation [66], and symbiotic bacteria supply nutrients to aphids, enhancing their adaptability to host plants [24,25]. However, the relationship between the microevolutionary processes of aphids related to the host plant and the symbiotic bacterial community remains unknown.

Phylogenetic analysis revealed distinct clustering of *A. odinae* samples from *Tr. sebifera* (Euphorbiaceae) and *R. chinensis* (Anacardiaceae) into separate groups (Figure 1). The topological structures indicated an evolutionary trend where the *Triadica*- and *Rhus*-Groups occupied a more evolved position, while samples from other host plants were situated at the base. Haplotype analysis further supported the specificity of Haplotype H2 and Haplotype H3 corresponding to these two host populations (Figure 2). The genetic distance indicated a 0.54% difference for the *Rhus*-Group (Haplotype H2) and a 0.18% difference for the *Triadica*-Group compared to the prevalent Haplotype H1, suggesting that these two host populations may be in the early stages of species differentiation, given the typical genetic distance threshold of around 2–2.5% between aphid species [67,68,69]. However, populations of *A. odinae* on other host plants did not exhibit significant differentiation.

Najar-Rodríguez et al. (2009) characterized bacterial symbionts associated with cotton aphids from Japan and Australia. They found that the bacterial composition of *A. gossypii* from Hibiscus and cotton was more similar to the bacterial composition of aphid species on other hosts in the same region than on the same plants in other regions [70]. In our study, a different finding was observed. Five aphid samples from different host plants within the same geographic area (Fuzhou, Fujian) were observed to share the same primary endosymbiont, *Buchnera*. However, their predominant secondary endosymbionts exhibited significant differences. Specifically, *Wolbachia* was present as the predominant secondary symbiont in the samples from *Tr. sebifera*, and *Serratia* was prevalent in the samples from *H. heptaphyllum* (Figure 4, Appendix A). Conversely, samples from other host plants rarely carried *Wolbachia* or *Serratia* as their predominant secondary symbionts. Notably, samples from Anacardiaceae (*R. chinensis*, *M. indica,* and *To. vernicifluum*) did not consistently carry any predominant secondary symbiont, except for HCX56.

In our analysis of phylogeny tree topologies and secondary symbiont compositions, a noteworthy pattern emerged among *A. odinae* populations on *Tr. sebifera*. These populations consistently formed a distinct clade, carrying *Wolbachia* as the predominant secondary symbiont. Two possible explanations arise: *Wolbachia* may contribute to *A. odinae* adaptation to *Tr. sebifera*, influencing their differentiation on this host plant; alternatively, *Wolbachia* might have become the predominant secondary symbiont during microevolution driven by host plants. Regardless, this finding suggests a potential connection between symbiotic bacteria communities and *A. odinae*’s adaptation to *Tr. sebifera.*

Interestingly, samples from *H. heptaphyllum* carried *Serratia* as their predominant secondary symbiont, suggesting a potential role in aiding *A. odinae*’s adaptation to this host. Despite this, no significant differentiation was observed on phylogenetic trees, hinting that changes in the symbiotic bacterial community might precede genetic differentiation in aphids. Conversely, previous studies on the Cucurbitaceae-(CU) and Malvaceae-specialized (MA) biotypes of *A. gossypii* showed high host specificity with no significant difference in symbiotic bacterial composition [71]. We found a similar mismatch in *A. odinae* samples from *R. chinensis*, which displayed clear differentiation on the phylogenetic tree, yet no predominant secondary symbiont was identified. The symbiotic bacterial community in these samples resembled those found on host plants from the Anacardiaceae family (*M. indica* and *To. vernicifluum*), implying changes in the symbiotic bacterial community likely occur after host-related differentiation.

This variety of connections between symbiotic bacterial communities and the microevolutionary process of *A. odinae* suggests a partial correspondence with host-related differentiation. The interplay between symbiotic bacteria and host plants may significantly contribute to the adaptation and differentiation of *A. odinae*. These findings underscore the intricate relationship between symbiotic bacteria and the evolution of aphids in response to their host plants, providing valuable insights for future studies on aphid adaptation and ecological interactions.

We also found something unexpected with two samples collected from *To. vernicifluum*. One sample (HCX56) harbored *Wolbachia* as the predominant secondary symbiont, resembling the *Triadica*-Group. Conversely, the other sample (HCX55) did not exhibit any predominant secondary symbiont, resembling samples collected from the Anacardiaceae family. Given that the analysis of phylogenetics indicated no significant genetic differentiation within the *A. odinae* population on *To. vernicifluum*, there is no evidence to rule out the possibility that samples collected from *To. vernicifluum* are not colonies of *A. odinae* from other host plants with different symbiotic bacterial community structures. Therefore, determining whether *A. odinae* feeding on *To. vernicifluum* can consistently carries *Wolbachia* as the predominant secondary symbiont requires further exploration. Moreover, the symbiotic bacteria carried by the samples from *M. indica* displayed a more diverse composition. We need to conduct further studies to determine if this diversity plays a role in the host-related differentiation of *A. odinae*. These findings open up new avenues for understanding the complex interactions between symbiotic bacteria and aphids on different host plants.

The alpha diversity analysis revealed notable variations in the number of species of symbiotic bacterial communities among *A. odinae* populations feeding on different host plants. This could be attributed to differences in the environmental microbial communities associated with their respective hosts, which could result in changes in the count of OTUs of non-symbiotic bacteria within the aphid’s bodies. There were no significant differences observed in species evenness; this may be due to the non-significant differences in the relative abundance of non-endosymbiotic bacteria and the primary endosymbiont (such as *Buchnera*) among aphids feeding on different hosts. However, both NMDS and PCoA analyses did not reveal distinct clustering of samples from different host plants, potentially attributed to the limited sample size and its inability to effectively mitigate systematic error. To gain more comprehensive insights into the variations and host-related differentiation of symbiotic bacterial communities in *A. odinae*, a more extensive dataset with a larger sample size from diverse host plants is essential. These findings suggest that the observed diversity patterns hold potential significance in understanding host-related differentiation within *A. odinae* and emphasize the need for further investigations with a more robust dataset to unravel the intricacies of symbiotic bacterial associations in this species.

## 5. Conclusions

Our study reveals significant genetic differentiation among aphid populations associated with different hosts, notably *Triadica sebifera* and *Rhus chinensis*. Additionally, the diversity analysis of endosymbionts in aphid samples from different hosts suggests variations in secondary symbiotic bacteria and endosymbiotic bacterial structures. These findings highlight the intricate interconnections among aphid microevolution, host plant adaptation, and symbiotic bacteria, providing valuable insights into the microevolutionary processes that shape the macroevolutionary relationships between bacterial symbionts and insects. Future research based on omics data is needed to determine the relative role of aphid itself and symbiotic bacteria in host plant adaptation.

## Figures and Tables

**Figure 1 animals-14-00283-f001:**
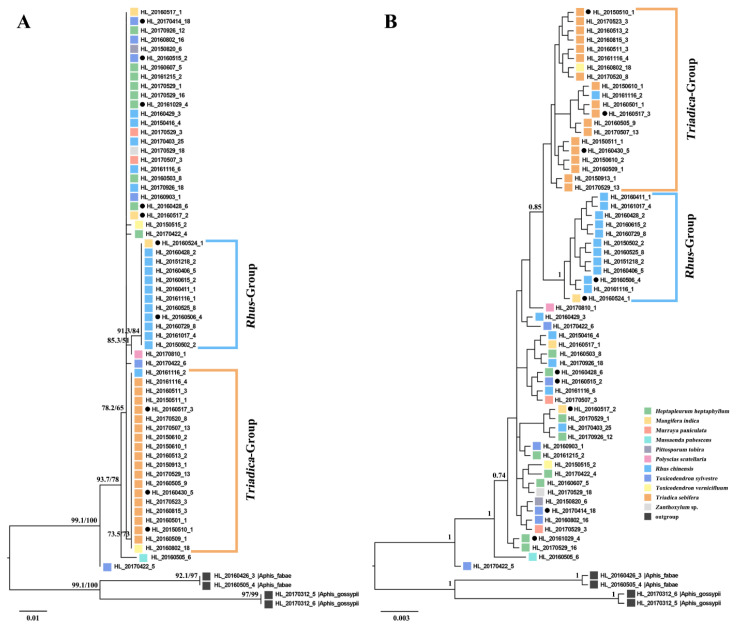
Phylogenetic trees of *Aphis odinae* populations in different host plants inferred from maximum likelihood (ML) (**A**) and Bayesian (BI) (**B**) analyses based on *COI*. Samples from different host plants are represented by different colors. The support values (UFBoot/SH-aLRT) and posterior probabilities (>0.7) are shown for main nodes. Solid circles mark the samples used for symbiotic bacterial diversity analysis.

**Figure 2 animals-14-00283-f002:**
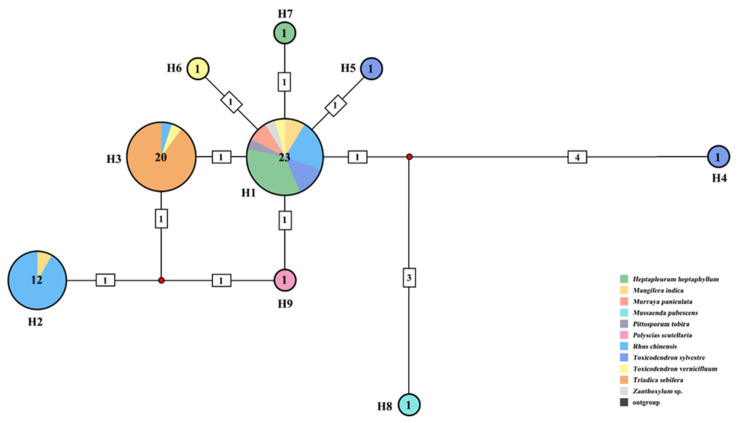
Haplotype network of *Aphis odinae* population based on *COI*. The colors in the circles represent different host plants, and the numbers by the sides of circles indicate haplotype numbers. Sample number of each haplotype is annotated in the center of each circle.

**Figure 3 animals-14-00283-f003:**
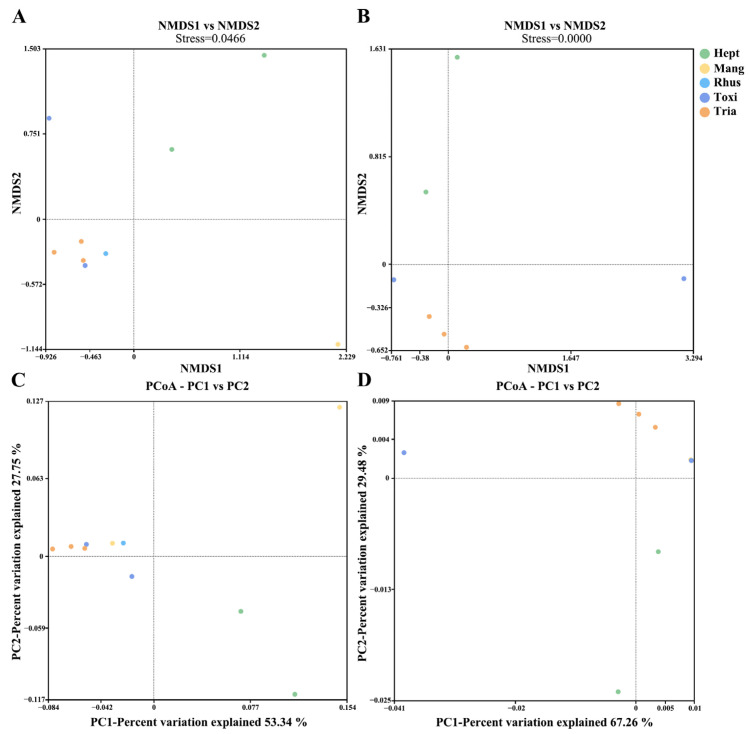
Nonmetric multidimensional scaling (NMDS) plots showing the beta diversity of the bacterial communities of *A. odinae* on different host plants based on Bray–Curtis (**A**) and Weighted Unifrac distances (**B**). Principal coordinate analyses (PCoA) plots illustrate the separation of samples based on Bray–Curtis (**C**) and Weighted Unifrac distances (**D**). Colors correspond to different groups, as shown in the legend; see Table 2 for detailed sample information.

**Figure 4 animals-14-00283-f004:**
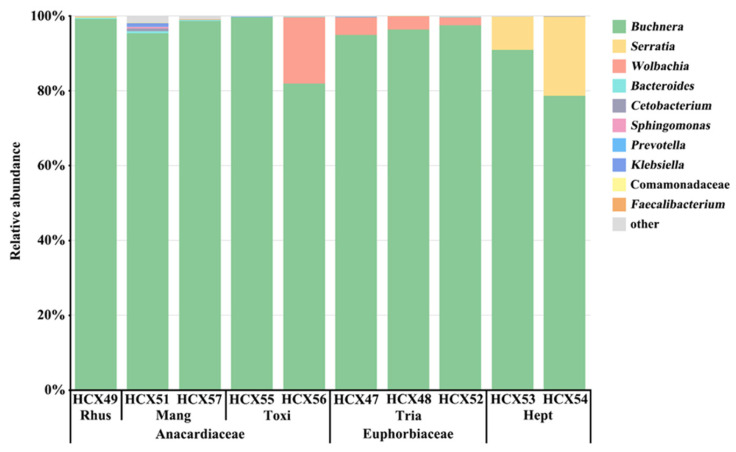
Taxonomic composition and relative abundance of symbiotic bacteria with ten *A. odinae* samples from different host plants. See Table 2 for detailed sample information.

**Table 1 animals-14-00283-t001:** *Aphis odinae* samples used in this study and the information of the host plants.

Sample ID	Host Plant	Host Family	Collection Locality
HL_20150416_4	*Rhus chinensis*	Anacardiaceae	Fuzhou, Fujian
HL_20150502_2	*Rhus chinensis*	Anacardiaceae	Quanzhou, Fujian
HL_20150510_1	*Triadica sebifera*	Euphorbiaceae	Fuzhou, Fujian
HL_20150511_1	*Triadica sebifera*	Euphorbiaceae	Fuzhou, Fujian
HL_20150515_2	*Toxicodendron vernicifluum*	Anacardiaceae	Fuzhou, Fujian
HL_20150610_1	*Triadica sebifera*	Euphorbiaceae	Fuzhou, Fujian
HL_20150610_2	*Triadica sebifera*	Euphorbiaceae	Fuzhou, Fujian
HL_20150820_6	*Pittosporum tobira*	Pittosporaceae	Meishan, Sichuan
HL_20150913_1	*Triadica sebifera*	Euphorbiaceae	Fuzhou, Fujian
HL_20151218_2	*Rhus chinensis*	Anacardiaceae	Fuzhou, Fujian
HL_20160406_5	*Rhus chinensis*	Anacardiaceae	Fuzhou, Fujian
HL_20160411_1	*Rhus chinensis*	Anacardiaceae	Fuzhou, Fujian
HL_20160428_2	*Rhus chinensis*	Anacardiaceae	-
HL_20160428_6	*Heptapleurum heptaphyllum*	Araliaceae	-
HL_20160429_3	*Rhus chinensis*	Anacardiaceae	Fuzhou, Fujian
HL_20160430_5	*Triadica sebifera*	Euphorbiaceae	Fuzhou, Fujian
HL_20160501_1	*Triadica sebifera*	Euphorbiaceae	Fuzhou, Fujian
HL_20160503_8	*Heptapleurum heptaphyllum*	Araliaceae	Fuzhou, Fujian
HL_20160505_9	*Triadica sebifera*	Euphorbiaceae	Fuzhou, Fujian
HL_20160506_4	*Rhus chinensis*	Anacardiaceae	Fuzhou, Fujian
HL_20160509_1	*Triadica sebifera*	Euphorbiaceae	Fuzhou, Fujian
HL_20160511_3	*Triadica sebifera*	Euphorbiaceae	Fuzhou, Fujian
HL_20160513_2	*Triadica sebifera*	Euphorbiaceae	Fuzhou, Fujian
HL_20160515_2	*Triadica sebifera*	Euphorbiaceae	Fuzhou, Fujian
HL_20160517_1	*Mangifera indica*	Anacardiaceae	-
HL_20160517_2	*Mangifera indica*	Anacardiaceae	-
HL_20160517_3	*Triadica sebifera*	Euphorbiaceae	-
HL_20160524_1	*Mangifera indica*	Anacardiaceae	-
HL_20160525_8	*Rhus chinensis*	Anacardiaceae	Fuzhou, Fujian
HL_20160607_5	*Heptapleurum heptaphyllum*	Araliaceae	Fuzhou, Fujian
HL_20160615_2	*Toxicodendron sylvestre*	Anacardiaceae	Fuzhou, Fujian
HL_20160729_8	*Rhus chinensis*	Anacardiaceae	Wuyishan, Fujian
HL_20160802_16	*Toxicodendron sylvestre*	Anacardiaceae	Wuyishan, Fujian
HL_20160802_18	*Toxicodendron vernicifluum*	Anacardiaceae	Wuyishan, Fujian
HL_20160815_3	*Triadica sebifera*	Euphorbiaceae	Linhai, Zhejiang
HL_20160903_1	*Toxicodendron sylvestre*	Anacardiaceae	Fuzhou, Fujian
HL_20161017_4	*Rhus chinensis*	Anacardiaceae	Fuzhou, Fujian
HL_20161029_4	*Heptapleurum heptaphyllum*	Araliaceae	Fuzhou, Fujian
HL_20161116_1	*Rhus chinensis*	Anacardiaceae	Fuzhou, Fujian
HL_20161116_2	*Rhus chinensis*	Anacardiaceae	Fuzhou, Fujian
HL_20161116_4	*Triadica sebifera*	Euphorbiaceae	Fuzhou, Fujian
HL_20161116_6	*Rhus chinensis*	Anacardiaceae	Fuzhou, Fujian
HL_20161215_2	*Heptapleurum heptaphyllum*	Araliaceae	-
HL_20170403_25	*Rhus chinensis*	Anacardiaceae	Fuding, Fujian
HL_20170414_18	*Toxicodendron sylvestre*	Anacardiaceae	Fuzhou, Fujian
HL_20170422_4	*Heptapleurum heptaphyllum*	Araliaceae	Fuzhou, Fujian
HL_20170422_6	*Toxicodendron sylvestre*	Anacardiaceae	Fuzhou, Fujian
HL_20170507_13	*Triadica sebifera*	Euphorbiaceae	Fuzhou, Fujian
HL_20170507_3	*Murraya paniculata*	Rutaceae	Fuzhou, Fujian
HL_20170520_8	*Triadica sebifera*	Euphorbiaceae	Fuzhou, Fujian
HL_20170523_3	*Triadica sebifera*	Euphorbiaceae	Fuzhou, Fujian
HL_20170529_1	*Heptapleurum heptaphyllum*	Araliaceae	Fuzhou, Fujian
HL_20170529_13	*Triadica sebifera*	Euphorbiaceae	Fuzhou, Fujian
HL_20170529_16	*Heptapleurum heptaphyllum*	Araliaceae	Fuzhou, Fujian
HL_20170529_18	*Zanthoxylum* sp.	Rutaceae	Fuzhou, Fujian
HL_20170529_3	*Murraya paniculata*	Rutaceae	Fuzhou, Fujian
HL_20170926_12	*Heptapleurum heptaphyllum*	Araliaceae	Fuzhou, Fujian
HL_20170926_18	*Rhus chinensis*	Anacardiaceae	Fuzhou, Fujian

**Table 2 animals-14-00283-t002:** Sample collection and grouping information for symbiotic bacterial diversity analysis of *A. odinae*.

Sample ID	Treat ID	Sequenced Region	Host Plant	Host Family	Host Group	Collection Locality
HL_20150510_1	HCX47	16s, v3+v4_b	*Triadica sebifera*	Euphorbiaceae	Tria	Fuzhou, Fujian
HL_20160430_5	HCX48	16s, v3+v4_b	*Triadica sebifera*	Euphorbiaceae	Tria	Fuzhou, Fujian
HL_20160506_4	HCX49	16s, v3+v4_b	*Rhus chinensis*	Anacardiaceae	Rhus	Fuzhou, Fujian
HL_20160517_2	HCX51	16s, v3+v4_b	*Mangifera indica*	Anacardiaceae	Mang	Fuzhou, Fujian
HL_20160517_3	HCX52	16s, v3+v4_b	*Triadica sebifera*	Euphorbiaceae	Tria	Fuzhou, Fujian
HL_20161029_4	HCX53	16s, v3+v4_b	*Heptapleurum heptaphyllum*	Araliaceae	Hept	Fuzhou, Fujian
HL_20160428_6	HCX54	16s, v3+v4_b	*Heptapleurum heptaphyllum*	Araliaceae	Hept	Fuzhou, Fujian
HL_20170414_18	HCX55	16s, v3+v4_b	*Toxicodendron sylvestre*	Anacardiaceae	Toxi	Fuzhou, Fujian
HL_20160515_2	HCX56	16s, v3+v4_b	*Toxicodendron sylvestre*	Anacardiaceae	Toxi	Fuzhou, Fujian
HL_20160524_1	HCX57	16s, v3+v4_b	*Mangifera indica*	Anacardiaceae	Mang	Fuzhou, Fujian

Note: Treat ID: the code name of samples used for symbiotic bacterial diversity analysis (HCX: the Chinese acronym for “metagenomic sequencing”); the samples are grouped according to the host plants; the group names are abbreviations of the genus name of host plants (Tria: *Triadica sebifera*; Mang: *Mangifera indica*; Rhus: *Rhus chinensis*; Hept: *Heptapleurum heptaphyllum*; Toxi: *Toxicodendron sylvestre*).

**Table 3 animals-14-00283-t003:** Results of ANOVA based on alpha diversity indices in bacterial symbiont communities of *Aphis odinae* in different host plants.

Alpha Diversity Index	Df	SS	MS	F	*p*
ACE	4	804.55	201.14	8.18	*0.02*
Chao1	4	672.81	168.20	29.10	*0.00*

Note: Df: between-groups degree of freedom; SS: sum of squares; MS: mean sum of squares.

## Data Availability

The data presented in this study are available in the article and Appendix A.

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
