# Peer review of "Partial Correspondence between Host Plant-Related Differentiation and Symbiotic Bacterial Community in a Polyphagous Insect"

_animals, 2024, doi:10.3390/ani14020283_

Round 1

Reviewer 1 Report

Comments and Suggestions for Authors

This paper analyzed the COI gene sequences of Aphis odinae collected from different host plants and construct a phylogenetic tree according to the COI gene sequences; also analyzed the diversity of bacterial endosymbionts in aphids collected from different host plants. The results showed that the aphids from different host plants showed a tendency to group according to the host plants, and the bacterial endosymbionts also varied with the host plants. This paper has certain significance for the study of the host adaptation of this aphid. The quality of the manuscript is good, and the quality of writing is also good.

Major concerns:

Aphis odinae in Fujian should be strictly parthenogenetic because the winter there is not very cold. For strictly parthenogenetic aphid, whether they will group according to the host plants depends largely on the founders. Although this paper found a tendency to group according to the host plants in a limited number of samples, it is not enough to say there is a structure by host plant because there are not samples out of Fujian. Bacterial endosymbionts are one of the factors affecting the host species of aphids (suchida et al., 2004), but it is not certain whether they are common in aphids. It is difficult to answer whether the difference in host adaptation is caused by the difference in aphids infected with different secondary symbiotic bacteria or because the difference in host adaptation is caused by the difference in secondary symbiotic bacteria, unless quantitative validation tests are carried out, which this work obviously did not do.

Minor concerns or suggestions:

1.       Why did the authors choose these host plants? Is there any other host plant for this aphid besides the hosts mentioned in the manuscript?

2.       Table 4. Why df=4? Is it divided into 4 treatments? Why is it divided into 4 treatments?

3.       Table 3. What does HCX47 stand for? Please give explanation in the table notes.

4.       The contents of Figure 4 and Table 3 are duplicated. I suggest keeping either one of them in the main text and the other as an appendix.

The following two papers show that the aphid host plant has no apparent relationship with the species of endophytic bacteria, and I suggest the authors to cite them in the Discussion.

Najar-Rodríguez, A.J.; McGraw, E.A.; Mensah, R.K.; Pittman, G.W.; Walter, G.H. The microbial flora of Aphis gossypii: Patterns across host plants and geographical space. J. Invertebr. 2009, 100, 123–126.

Guo H, Yang F, Meng M, Feng J, Yang Q, Wang Y. No Evidence of Bacterial Symbionts Influencing Host Specificity in Aphis gossypii Glover (Hemiptera: Aphididae). Insects. 2022 May 14;13(5):462. doi: 10.3390/insects13050462.

Reviewer 2 Report

Comments and Suggestions for Authors

Title:

ok.

Abstract

Highlight Novelty (Line 33-34):

Conclude by summarizing the study's novelty or contribution.

Introduction

The transition between Paragraphs:

Consider improving the transition between the paragraphs discussing aphid-bacteria associations and the specific focus on Aphis odinae. A smoother transition will help readers follow the logical flow of your introduction.

Clarification on Research Gap:

In the last sentence, it mentions that "most of the research did not further analyze the population differentiation of aphids on different host plants," but it would be helpful to explicitly state the specific gap in the literature that your study aims to address.

Check Grammar and Syntax:

In some places, sentence structures are complex. Ensure that the wording is clear and grammatically correct. For instance, in the sentence "Whether ongoing differentiation exists and whether the degrees of differentiation are associated with different symbiotic bacterial communities," consider breaking it into two sentences for clarity.

Materials and Methods

Sample Collection (Lines 82-89):

Clarify criteria for host plant and location selection. (Lines 84-85, 91-92)

Storage Conditions (Lines 89-90):

Explain the rationale for -20 ℃ storage for aphid samples. (Line 89)

16S rRNA Gene Sequencing (Lines 114-153):

Explain choice of V3-V4 region for sequencing. (Line 146)

Data Analysis (Lines 153-188):

Specify criteria for quality filtering in data analysis. (Lines 156-158)

Consistency (Lines 188-188):

Maintain consistent terminology throughout. (Lines 82-89, 92-153)

Results

Presented Well

Discussion:

Clarity of Phylogenetic Analysis (Line 283-293):

The discussion of clades and groups formed in the phylogenetic tree could be clearer. Consider simplifying the language and providing a concise summary of the major findings to enhance accessibility for a broader audience.

Secondary Symbionts (Line 306-315):

The discussion on secondary symbionts is informative but somewhat detailed. Streamline the presentation by focusing on key findings and their implications. Ensure that the importance of these findings in the context of the study is clearly conveyed.

Hypotheses Regarding Symbiotic Bacteria (Line 316-328):

The hypotheses proposed are intriguing, but it would be beneficial to present them in a more structured manner. Clearly state each hypothesis and its potential implications. This will aid readers in understanding the logical flow of your arguments.

Connect Alpha Diversity Analysis to Main Findings (Line 344-351):

While discussing alpha diversity analysis, connect these findings back to the main research question. Clarify the implications of these diversity patterns in the context of host-related differentiation in A. odinae.

Conclusion Summary (Line 353-362):

The conclusions are well-organized. Consider summarizing the main findings more explicitly, emphasizing their significance, and potentially suggesting how they contribute to the broader understanding of aphid microevolution.

Final Remarks (Line 363-364):

The final sentences are insightful. Consider adding a closing sentence that explicitly states the overarching significance of these findings, potentially in the context of future research or applications.

General comments

 After carefully reviewing the manuscript, I must commend the author for their skillful writing and overall presentation. However, I have identified several areas where the manuscript could be improved. These suggestions will help the author further enhance the manuscript's readability, structure, and impact.

Comments on the Quality of English Language

Careful proofreading is required. 
